# Rationalisation of Antifungal Properties of α-Helical Pore-Forming Peptide, Mastoparan B

**DOI:** 10.3390/molecules27041438

**Published:** 2022-02-21

**Authors:** Edward Jianyang Lim, Eunice Goh Tze Leng, Nhan Dai Thien Tram, Mercy Halleluyah Periayah, Pui Lai Rachel Ee, Timothy Mark Sebastian Barkham, Zhi Sheng Poh, Navin Kumar Verma, Rajamani Lakshminarayanan

**Affiliations:** 1Ocular Infections and Anti-Microbials Research Group, Singapore Eye Research Institute, The Academia, 20 College Road, Discovery Tower, Singapore 169856, Singapore; e0012314@u.nus.edu (E.J.L.); eunice.goh.t.l@seri.com.sg (E.G.T.L.); mercy.halleluyah.periayah@seri.com.sg (M.H.P.); 2Department of Pharmacy, National University of Singapore, 18 Science Drive 4, Singapore 117543, Singapore; nhan.tram@u.nus.edu (N.D.T.T.); phaeplr@nus.edu.sg (P.L.R.E.); 3Department of Laboratory Medicine, Tan Tock Seng Hospital, 11 Jalan Tan Tock Seng, Singapore 308433, Singapore; timothy_barkham@ttsh.com.sg; 4Lee Kong Chian School of Medicine, Nanyang Technological University Singapore, Clinical Sciences Building, 11 Mandalay Road, Singapore 308232, Singapore; zhisheng001@e.ntu.edu.sg; 5National Skin Centre, 1 Mandalay Road, Singapore 308205, Singapore; 6Academic Clinical Program in Ophthalmology and Visual Sciences Academic Clinical Program, Duke-NUS Medical School, Singapore 169857, Singapore

**Keywords:** antifungal peptides, drug resistance, invasive fungal infections, non-albicans *Candida*, skin wounds

## Abstract

The high mortality associated with invasive fungal infections, narrow spectrum of available antifungals, and increasing evolution of antifungal resistance necessitate the development of alternative therapies. Host defense peptides are regarded as the first line of defense against microbial invasion in both vertebrates and invertebrates. In this work, we investigated the effectiveness of four naturally occurring pore-forming antimicrobial peptides (melittin, magainin 2, cecropin A, and mastoparan B) against a panel of clinically relevant pathogens, including *Candida albicans*, *Candida parapsilosis*, *Candida tropicalis*, and *Candida glabrata*. We present data on the antifungal activities of the four pore-forming peptides, assessed with descriptive statistics, and their cytocompatibility with cultured human cells. Among the four peptides, mastoparan B (MB) displayed potent antifungal activity, whereas cecropin A was the least potent. We show that MB susceptibility of phylogenetically distant non-candida albicans can vary and be described by different intrinsic physicochemical parameters of pore-forming α-helical peptides. These findings have potential therapeutic implications for the design and development of safe antifungal peptide-based drugs.

## 1. Introduction

Invasive candidiasis (IC) is an infection caused by various *Candida* species, which results from candidemia and deep-seated tissue candidiasis. IC is observed among healthcare-associated infections in developed countries and is associated with high mortality rates (~40%) despite the institution of antifungal therapy [1]. Annually, the infection affects >250,000 people globally and is responsible for >50,000 deaths [2]. Candidemia, defined by the presence of *Candida* in the bloodstream, may cause infection of distant organs such as the liver or spleen, which may lead to inflammation and subsequent organ failure [2]. The incident rates of candidemia are higher among ageing populations and often comorbid with malignancy. 

*Candida albicans* is the major etiological agent in candidemia in the past. The global shift in favor of non-albicans such as *C. parapsilosis* and *C. tropicalis* in Asia, Southern Europe, and South America and *C. glabrata* in north America and northern Europe is troubling, owing to their heightened resistance to antifungal therapy [3,4]. Non-albicans *Candida* (NAC) strains represent more than 50% of the blood stream isolates, whereas *C. albicans* is the most common in clinical settings [5,6]. Acquired resistance to echinocandins and azole antifungals has been over-represented by non-albicans such as *C. glabrata* and *C. parapsilosis* [7]. The limited availability of antifungal agents plus the intrinsic resistance of fungal pathogens to the available antifungals complicates the treatment modalities of antifungal therapy. In addition, the development of new antifungals is more challenging than antibacterials since the antifungal drug targets have high similarity to human genes and gene products, thus having the potential for interaction with host tissue, leading to toxicity in the therapeutic regimen [8]. 

Antimicrobial peptides (AMPs) are ancient armors and form an important component of natural innate immunity found in various organisms [9]. AMPs are being developed as novel antimicrobial agents due to their broad-spectrum effects against bacteria, fungi, and viruses, as well as their lower likelihood to cause resistance in micro-organisms compared to traditional antibiotics [10,11]. In vitro and in vivo proof-of-concept studies have indicated that cationic host defense peptides (HDPs) have potential for clinical development owing to their antimicrobial and anti-inflammatory properties [12]. 

Membrane permeabilizing peptides have long been described as a possible novel class of antibiotics against multidrug-resistant pathogens [13,14,15]. Microbes are less likely to develop resistance towards these peptides due to the heightened fitness cost for microbes to modify the entire membrane. Melittin (ML), magainin 2 (MG), cecropin A (CA), and mastoparan B (MB) are a group of amphipathic cationic α-helical pore-forming peptides that were shown to interact with natural and artificial membranes, thus possessing high antimicrobial and varying cytotolytic properties despite their differences in length, charge, and specificity [16]. ML is a 24-residue linear cationic peptide and is a major component of the European honeybee *Apis mellifera*. MB is composed of 14 amino acid residues isolated from hornet venom *Vespa basalis*. Cecropin A is a 37-residue antimicrobial peptide isolated from hemolymph of *Hyalophora cecropia* and magainin 2 is a 23-residue cationic peptide isolated from skin of African clawed frog *Xenopus laevis*. 

Pore-forming peptides represent the most common AMPs, which are characterized by broad-spectrum antimicrobial, anti-inflammatory, anti-cancerous, and antioxidant properties [17,18]. However, their effectiveness against pathogenic and antifungal resistant NACs remains less understood. In this study, we aimed to investigate the physicochemical properties of MB, MG, ML, and CA, and their contribution to the antifungal activity against a panel of drug-resistant NAC strains. The cytocompatibility of the peptides was determined using cultured human keratinocytes and dermal fibroblasts. Finally, we investigated the mechanism of antifungal activity of the most potent pore-forming peptide.

## 2. Materials and Methods

### 2.1. Peptides and Cell Lines

All the peptides used in this study were purchased from M/s Mimotopes Pte Ltd., Clayton Victoria, Australia. The homogeneity of the peptides was confirmed by reversed phase high performance chromatography (RP-HPLC) and mass spectrometry by the supplier. RP-HPLC profile of purified peptides as received is shown in the Appendix A. The lyophilized peptides were reconstituted in appropriate buffers/media for determining their biological activity. Human immortalized keratinocytes (HaCaT) were from AddexBio (San Diego, CA, USA; T0020001), and NTLG1 cells, originally from the immortalized primary human foreskin keratinocyte cell line N/TERT as described previously [19], were kindly provided by Professor Maurice Adrianus Monique van Steensel, Nanyang Technological University Singapore. Normal adult human dermal fibroblasts (HDFs) were from American Type Culture Collection (ATCC, Manassas, VA, USA; PCS-201-012). All cell lines underwent routine Mycoplasma testing with GeneCopoeia MycoGuard™.

### 2.2. Circular Dichroism (CD) Spectropolarimetry

Far UV–CD spectra of the peptides in phosphate-buffered saline (PBS) or trifluoroethanol (TFE) were recorded on a Chirascan CD Spectrometer (Applied Photophysics Ltd, Leatherhead, UK) using a 0.1 cm path length quartz cuvette at 20 °C. Spectra were recorded from 190 nm to 260 nm at a 1.0 nm interval with a scan rate of 50 nm/min, and the final spectrum was taken to be the average of three scans. The helical content was estimated, following the protocol reported before [20]. 

### 2.3. Zeta-Potential (ζ) Measurements

Measurements of the *ζ*-potential of the peptides were performed using Litesizer^TM^ 500 (Anton Paar, Graz, Austria). Peptide samples were freshly prepared at 100 μg/mL in PBS, which was pre-filtered using a 0.22 μm membrane. Samples were transferred to an omega cuvette for measurements at 25°. The *ζ*-potential in mV was automatically calculated by the Litesizer^TM^ 500 (Anton Paar GmbH, Graz, Austria) using the Smoluchowski equation:*ζ* = (4π*ημ*)/*ε*

where *μ* is the electrophoretic mobility measured by the Litesizer ^TM^ 500, *η* is the viscosity of PBS, and *ε* is the dielectric constant of PBS.

The experiments were repeated three times, and the results were expressed as mean ± standard deviation.

### 2.4. Calculation of Physicochemical Parameters

Hydrophobicity (H) and hydrophobic moment (μH) of peptides were obtained from the Heliquest” web server (http://heliquest.ipmc.cnrs.fr/, accessed on 12 July 2021) [21].

### 2.5. Determination of the Minimum Inhibitory Concentrations (MICs)

MICs of the test peptides were determined using the broth microdilution method in 96-well microtitre plates. Briefly, 100 μL of two-fold serial dilutions of each peptide (adjusted to a final concentration of 1–620 μM) were transferred to the microtitre plates in duplicate. The test strains were cultured in Sabouraud Dextrose Agar (SDA) plates incubated for 24 h at 37 °C. Inoculums of the fungal strains were prepared by selecting isolated fungal colonies from subcultures and suspending the colonies in Sabouraud Dextrose Broth (SDB). The suspension concentration was adjusted to the desired final concentration of 10^4^–10^5^ CFU/mL. Final suspension (100 μL) was added to each well of the 96-well plates and incubated at 35 °C for 24 h. The optical density (OD) at 600 nm (OD_600_) was determined using a TECAN Infinite 200 microplate reader (Mannedorf, Switzerland). The MIC was taken to be the lowest peptide concentration which showed no visible microbial growth as well as by measuring the absorbance OD_600_. 

### 2.6. Mammalian Cell Viability Assays

Cell viability was determined using CellTiter 96^®^ Aqueous One solution cell proliferation assay kit (Promega, Madison, USA), according to the supplier’s instructions. HaCaT and HDF cells were cultured in Dulbecco’s Modified Eagle Medium (DMEM) supplemented with fetal bovine serum 10% (*v*/*v*) and antibiotics (50 U/mL penicillin and 50 μg/mL streptomycin). Cells were seeded in 96-well tissue culture plates at a cell density of 4.5 × 10^3^ cells/well and incubated at 37 °C for 24 h. The cells were then treated with a range of peptide concentrations (2–600 μM) obtained from serial dilution in the cell culture medium and the plates containing treated cells were incubated again at 37 °C for another 24 h. MTS solution (20 μL, provided in the assay kit) was next added to each well followed by incubation at 37 °C for 2 h. Absorbance was measured at 490 nm using a TECAN microplate reader and normalized with positive and negative control values to obtain % cell viability. In another set of experiment, the peptide treated cells were fixed with formaldehyde (3.7%, *v*/*v*) at room temperature for 15 min. After fixation, the cells were first washed with PBS, then Triton-X-100 (0.3%, *v*/*v*) was added into each well to permeabilize the cells. The cells were washed again with PBS and the adherent cells were stained with fluorescein isothiocyanate labelled anti-α-tubulin antibody (green), rhodamine phalloidin (red) and Hoechst 33342 (blue) to visualize cytoskeleton and nuclei morphologies. Cell images were acquired by an automated microscope IN Cell Analyzer 2200 (Model 2200, GE Healthcare UK Limited, Buckinghamshire, UK) equipped with the IN Cell Investigator software.

### 2.7. Cell Migration Assay

NTLG1 cells were grown to confluency in keratinocyte serum-free medium in a 96-well plate. Identical scratches were created using AccuWound 96 Scratch Tool (Agilent Technologies Inc., Santa Clara, CA, USA) and then washed in PBS before being replenished with fresh medium containing peptides. Wound closure was monitored by EVOS M5000 (Thermofisher Scientific, Singapore) and subsequently quantified by ImageJ to determine the effect of peptide relative to untreated control.

### 2.8. Sytox™ Green Uptake Assay

Overnight cultured fungal colonies were suspended in 16% SDB in PBS and adjusted to obtain an OD of 0.4 at 600 nm. A 5 mM Sytox™ green stock solution in dimethylformamide was added to the suspension to a final concentration of 1 μM and incubated for 15 min at room temperature in dark. The suspension (600 μL) was then added to a quartz cuvette and placed inside the Quanta Master spectrofluorometer. Once a constant plateau was reached, peptide solution (0.5×–4× MIC) was added to the cuvette. The increase in emission intensity at 515 nm was monitored after the addition of various concentrations of the peptides (λ_ex_ = 485 nm).

### 2.9. Time–Kill Kinetics Studies (TKS)

TKS was performed on *C. parapsilosis* 1002 (1.2 × 10^4^ CFU/mL) and *C. glabrata* 1003 (3.4 × 10^4^ CFU/mL) inoculum and exposed to MB peptide at 0.5×–4× MIC. At various time intervals, 100 μL of aliquots serial dilutions were plated on SDA plates and incubated at 35 °C for 48 h, and then the CFUs were counted and expressed as log_10_ CFU/mL. PBS-exposed strains were included as an untreated control. To determine the fungicidal activity of MB, *C. parapsilosis* 1002 strains (~1.5–2.0 × 10^5^ CFU/mL) in SDB were exposed to the peptide (9.92 μM) for 6 h. Viable cells were enumerated as above after 48 h. To determine the effect of monovalent or divalent ions, the broth was adjusted to appropriate concentrations of the ions.

## 3. Results

### 3.1. Physicochemical Properties of Pore-Forming Peptides

We compared the hydrophobicity, overall net charge, and isoelectric points of the four pore-forming peptides—MB, MG, MC, and CA (Table 1). These parameters are the key determinants of solubility and aggregation status (pI), interaction with acidic lipids (net charge), and membrane discrimination. The results suggest that MB is the most hydrophobic and MG is the least cationic among the four peptides (Table 1). Zeta-potential measurements further confirmed that MG had the lowest ζ-potential value, whereas ML had the highest value among these peptides. To determine the secondary structures of the peptides, we recorded their CD spectra in PBS and in a membrane-mimetic solvent (30% TFE). All the peptides displayed an unordered secondary structure in PBS, whereas a clear transition to α-helical structure was observed in TFE (Appendix A). Quantitative estimation of the α-helical content suggested that CA had the lowest % α-helicity whereas ML displayed the highest helical content in TFE. The helical content was comparable for MB and MG peptides (Table 1).

### 3.2. Antifungal Activity of Pore-Forming Peptides

We determined the MIC values of the peptides against 35 strains of “*Candida* spp.”, covering both albicans and NACs strains (Table 2). The different groups of *Candida* strains displayed varying degrees of susceptibility to the antimicrobial peptides. All the peptides were potent against the tested *C. albicans* strains. MB, ML, and CA were effective with a median MIC of 1.26, 1.41, and 2.00 μM, respectively, whereas MG required higher concentration (median MIC 13.24 μM) for complete inhibition. Among the non-albicans strains, *C. tropicalis* strains were susceptible to all the four pore-forming peptides with median MIC ranging from 0.62 μM (MB) to 4.22 μM (ML). *C. parapsilosis* strains displayed resistance to CA as no inhibitory effects were observed even at the highest concentration tested but were susceptible to other pore-forming peptides. The median MICs were 4.96 μM, 5.62 μM, and 13.28 μM for MB, ML, and MG, respectively. Among the fungal species, *C. glabrata* displayed heightened resistance to ML as MIC values could be determined against three clinical isolates only. At least 7 out of 10 *C. glabrata* strains were susceptible to MB and MG with a median MIC of 4.96 μM and 26.48 μM, respectively. However, none of the *C. glabrata* strains were susceptible to CA. 

### 3.3. Host Cell Cytocompatibility of Pore-Forming Peptides

Next, we determined the cytotoxicity of peptides for HaCaT and HDF cells. The dose-dependent changes in the metabolic activity and cell morphology were investigated by MTS-based assays and automated imaging, respectively. Among the four pore-forming peptides, ML displayed heightened toxicity for both the mammalian cell-types as a complete loss of viability was observed at one to two times the median MIC value of *C. albicans* (Figure 1c). Compared to ML, cytotoxic effects of MB were instead observed at elevated concentrations (60× median MIC value against *C. albicans*) (Figure 1a). MG and CA did not show any major cytotoxic effect for both cell-types at the range of concentrations tested (Figure 1b,d).

To obtain better insight into the status of the cell health, we performed the high-content analysis of the peptide treated cells (Figure 2). No significant change in the cellular morphology was observed for HaCaT cells exposed to 38.8 μM MB when compared to untreated cells. However, substantial disruption of the microcolonies and reduction in actin staining was observed at a MB peptide concentration of 77.5 μM. HaCaT cells exposed to 5.6 μM of ML appeared fragmented with a lower number of microcolonies compared to untreated cells, thus confirming the cytotoxic effects of the peptide. Cells treated with MG or CA retained the microcolonies with well-developed adhesion systems that appeared to be similar to the untreated control, indicating a lack of significant cytotoxic effects of the two pore-forming peptides. Similar concentration-dependent changes in the morphology were observed for HDFs treated with MB or ML, whereas MG and CA did not show any major signs of cytotoxicity for HDFs, thus corroborating the MTS results (Figure 2). 

One of the hallmarks of wound repair is the migration of cells to the injured sites. To infer whether the peptides alter the cell migration, we determined the effect of peptides on the migration of immortalized human keratinocytes by scratch wound assay (Figure 3a). The results suggested a significant decrease in the wound closure after treatment with MB (4.8 μM) or ML (2.8 μM) when compared to untreated control (Figure 3b). However, no difference was observed when treated with MG or CA peptides. Overall, these results indicated that MB was less cytotoxic than ML, while the other two pore-forming (MG and CA) peptides remained the least toxic for the skin cells, corroborating the MTS and high content analysis.

### 3.4. Antifungal Activity of MB in the Presence of Mono and Divalent Cations

Time–kill kinetics studies suggested that MB elicited rapid candidacidal activity against *C. parapsilosis*. There was a >2 log_10_ reduction in fungal viability within 30 min at 2× and 4× MIC (Figure 4a). A similar effect was observed against *C. glabrata* at 4× MIC (Figure 4b). To infer the pore-forming properties of MB, we determined ATP release of fungal strains after exposure to 4× MIC of MB. ATP release studies (data not shown) indicated that there was a complete loss of intracellular ATP within 6 h of exposure of *C. parpsilosis* to MB. To further discern the membrane perturbation by MB, we recorded the fluorescence influx of the membrane impermeable Sytox green dye upon addition of the peptide to fungal inoculum. MB elicited rapid and concentration-dependent increase in the uptake of Sytox green, indicating substantial membrane perturbation at 2× and 4× MIC values against both *C. parapsilosis* (Figure 4c) and *C. glabrata* (Figure 4d) strains. Thus, these results demonstrate that MB elicits rapid candidacidal properties by perturbing the cytoplasmic membrane and removing vital intracellular components. The fungicidal property of the peptide was dependent on the concentrations of monovalent (NaCl) or divalent (CaCl_2_) ions present in the broth (Table 3). The presence of a high concentration of Na^+^ or Ca^2+^ attenuated the fungicidal properties of MB. 

### 3.5. Correlation of Physicochemical Properties of MB with Anti-C. glabrata Activity

α-helical HDPs are well-characterized peptides for drug optimization by quantitative structure activity correlation [22,23,24,25,26]. Yount et al. reported that α-helical HDPs or their signature in mammalian proteome could be distinguished by hydrophobic moment, overall net charge, hydrophobicity, and lysine-to-arginine ratio [27]. These authors further demonstrated that the membrane permeation of α-helical HDPs could be predicated by integration of hydrophobic moment and charge. Among the four pore-forming peptides, MB displayed broad antifungal activities against both *C. albicans* as well as NACs. However, *C. parapsilosis* and *C. glabrata* strains displayed lower susceptibility to ML, MG, and CA when compared to *C. albicans* or *C. tropicalis* strains. Therefore, we hypothesized that the amino acid sequence of MB is optimized to confer potent antifungal activities against NACs. To understand the distinct properties of MB, we searched the PUBMED and Google Scholar using the key words such as “Antifungal peptides” and “Peptides and *C. glabrata*/*tropicalis*” and identified research articles that reported the MIC values of natural or synthetic peptides against the two pathogens. To determine the physicochemical parameter that better describes the antifungal properties of MB, we searched the literature that reported the MIC values of short α-helical peptides (≤18 residues). A total of 41 and 42 peptides that showed antifungal activity against *C. glabrata* and *C. tropicalis*, respectively, were identified from the search (Appendix A). We then computed their physicochemical parameters such as hydrophobicity (H), hydrophobic moment (μH), and overall net charge (Q) for these peptides.

The correlation between log_10_ MIC values and the various physicochemical parameters are shown in Figure 4 and Figure 5. The scattered data points around the lines of best fit are attributed to inter-laboratory variations and experimental conditions employed for the determination of MIC values. There was a statistically significant association between log_10_ MIC of the peptides that displayed potent activity against *C. glabrata* strains and Q (r = 0.53 and *p* = 3 × 10^−4^) (Figure 5c). There were no statistically significant associations (*p* > 0.05) observed between log_10_ MIC of *C. glabrata* peptides and H (Figure 5a) or μH (Figure 5b). However, the integration of the latter two parameters with Q resulted in statistically significant associations with the log_10_ MIC values of *C. glabrata* (Figure 5d,e). There was a statistically significant association between log_10_ MIC of the peptides that displayed potent activity against *C. tropicalis* strains and H (r = 0.38 and *p* = 1.3 × 10^−2^) (Figure 6a), as well as μH (r = 0.41 and *p* = 7.0 × 10^−3^) (Figure 6b). However, there was no statistically significant association between the log_10_MIC values of *C. tropicalis* and Q or integration of the parameters (i.e., H*Q or μH*Q, Figure 6c–e).

## 4. Discussion and Conclusions

Invasive fungal infections caused by candida species represent the fourth leading cause of hospital acquired infections with high mortality [28]. Although *C. albicans* is the major etiological agent in candidiasis, the increased prevalence of NACs has been reported worldwide. Among the NAC species, *C. glabrata* remains the most common while *C. parapsilosis*, *C. tropicalis*, and *C. krusei* are also frequently isolated [29]. The rise of antifungal resistance among clinically relevant Candida species and the differences in the pattern of antifungal resistance limit the choice of antifungal therapy. In this study, we investigated the antifungal properties of four prolific pore-forming peptides against *C. albicans* and NAC strains. Investigation of the physicochemical parameters and secondary structural analysis suggest that the peptides are amphipathic and cationic, as indicated by high pI and positive zeta-potential values. Among the peptides, ML had the maximum zeta-potential value when compared to other peptides, though CA had the highest overall net charge. Similarly, the helical content of the peptides in membrane-mimetic solvent was the highest for ML and the lowest for CA. 

All the four peptides displayed potent activity against *C. albicans* and *C. tropicalis* strains, though a higher MIC was observed for MG against *C. albicans*. Two *C. tropicalis* strains displayed higher MIC values against ML, MG, and CA peptides, but were susceptible to MB. Thus, MG was the least potent and MB was the most potent against the two strains. Both *C. parapsilosis* and *C. glabrata* were not susceptible to CA and showed enhanced resistance to ML. Memariani and Memariani compiled the antifungal properties of ML against a wide range of fungal species and indicated a more than 50 times increase in the MIC of ML against *C. glabrata* when compared to *C. albicans* [30], thus corroborating our in vitro assays. A total of 7 of the 10 *C. glabrata* strains were susceptible to MB peptide, whereas the other three strains displayed heightened resistance to the peptide. These in vitro results shed further light on the remarkable resistance of *C. glabrata* against prolific pore-forming peptides. A number of surveillance studies documented the enhanced resistance of *C. glabrata* against echinocandins, cyclic lipopeptides, underlining the need for alternative therapies for treating such infections [31,32,33]. The reduced susceptibility of *C. glabrata* to salivary proteins histatin-5, human-β-defensins 2 and 3, lactoferrin, sheep myeloid antimicrobial peptide, and other natural antimicrobial peptides has been reported. Phylogenetically, *C. parapsilosis* and *C. glabrata* are a distant lineage from *C. albicans,* and *C. glabrata* is more closely related to non-pathogenic *Saccharomyces cerevisiae*. Schmalreck et al. reported a linkage between azole susceptibility and phylogenetic relationship among 20 pathogenic fungal species covering nearly 10,000 yeast strains [34]. The observation that ML required a high concentration to elicit fungicidal activity against *S. cerevisiae* and *C. glabrata* augment the lineage specific susceptibility to pore-forming peptides [35]. 

The antifungal properties of ML and CA against *C. albicans* have been attributed to the interaction of the peptides with cytoplasmic membrane with a concomitant increase in membrane potential loss [36,37]. It has been shown that ML displayed lethal candidacidal properties through membrane disruption, and the lethal action was energy-independent [36]. We determined that ML was potent against various Gram-negative and Gram-positive bacteria at sub-μM concentration, but a higher concentration was required for the inhibition of *C. parapsilosis* and *C. glabrata strains*. The lack of activity is substantial against *C. glabrata* strains as no detectable MIC was observed within the tested concentration range. These results suggest the distinctive feature of the cell wall components of *C. glabrata* that would interfere with the interaction of peptides with cytoplasmic membrane [38]. 

All the four peptides displayed varying degree of cytotoxicity for skin cells, as determined by MTS and high-throughput screening. ML displayed heightened toxicity to both keratinocytes and fibroblasts followed by MB, whereas MG and CA remained non-cytotoxic, even having elevated concentration. In support of these observations, MG and CA did not affect the migration of keratinocytes, whereas a significant delay was observed for both MB and ML peptides, the latter causing the maximum delay in wound closure.

Among the various α-helical peptides, MB displayed the most potent antifungal activity against both *C. albicans* and NACs. Secondary structure analysis and zeta potential measurements indicated that ML had the highest α-helical content and zeta potential values. There was no clear correlation between the antifungal properties of α-helical peptides with the experimentally determined physicochemical parameter. In addition, MB displayed salt-concentration-dependent antifungal activity, a common trait associated with cationic antimicrobial peptides. Sytox green uptake assay indicated rapid uptake of the DNA-binding probe upon the addition of peptide to *C. glabrata* or *C. parapsilosis* strains, indicating that the peptide perturbs the cytoplasmic membrane of the yeast strains. Time–kill kinetics studies confirmed the substantial lethality associated with peptide-induced membrane perturbation. 

To determine the physicochemical properties that can describe the antifungal activity of MB, a comprehensive literature search on short α-helical peptides that reported the MIC values of the peptides against *C. glabrata* strains was carried out. For a comparison, we have included *C. tropicalis* strains in order to infer whether phylogenetically distant species share a common susceptibility pattern against α-helical peptides. The analysis indicated that the overall net charge of the peptides and H*Q or μH*Q could describe the anti-*C. glabrata* activity of these peptides. Similarly, H and μH of the peptides could describe the anti-*C. tropicalis* activity of peptides. The correlation between physicochemical parameters and antifungal susceptibility is striking considering that the above analysis was carried out among a diverse heterogenous group of peptides. Taken together, these observations suggest different physicochemical requirements for antifungal peptide susceptibility among phylogenetically different *Candida* spp. strains. The approach reported here enabled us to determine the key physicochemical parameters of MB. 

In conclusion, we compared the antifungal properties of prolific pore-forming antimicrobial peptides and their cytotoxicity for skin cells. Among the four pore-forming peptides, MB possessed physicochemical properties that can be fine-tuned to be effective against various *Candida* spp. strains.

## Figures and Tables

**Figure 1 molecules-27-01438-f001:**
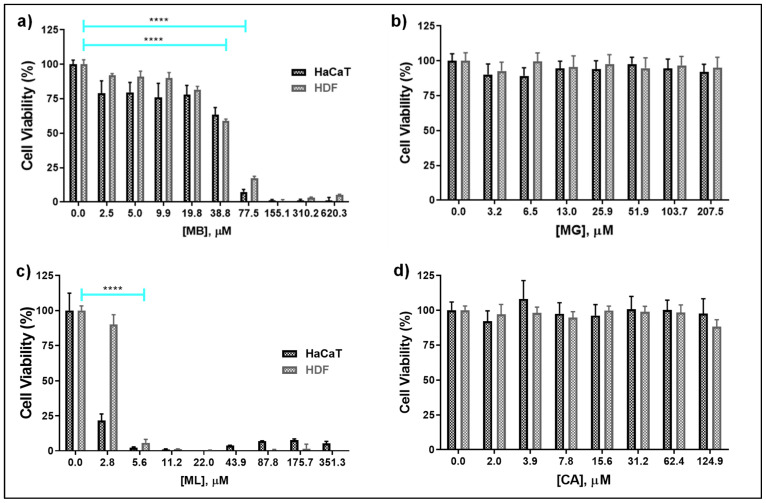
Cytotoxicity of pore-forming peptides for HaCaT and HDF cells, determined by MTS assay. (**a**) MB; (**b**) MG; (**c**) ML; (**d**) CA. Each bar represents mean ± s.d. from three independent triplicate experiments. Confluent cells were treated with increasing concentrations of indicated peptides. After 24 h, the metabolic activity of cells was quantified by MTS-based assay and the absorbance values, normalized in comparison to untreated control, were presented as cell viability (%). Note that MB decreased the metabolic activity of the cells at elevated concentrations (>38.8 µM) whereas heightened cytotoxicity was observed for cell treated with low concentrations of ML. **** *p* < 0.0001 compared to corresponding untreated (0.0) controls as determined by Dunnett’s multiple comparison test.

**Figure 2 molecules-27-01438-f002:**
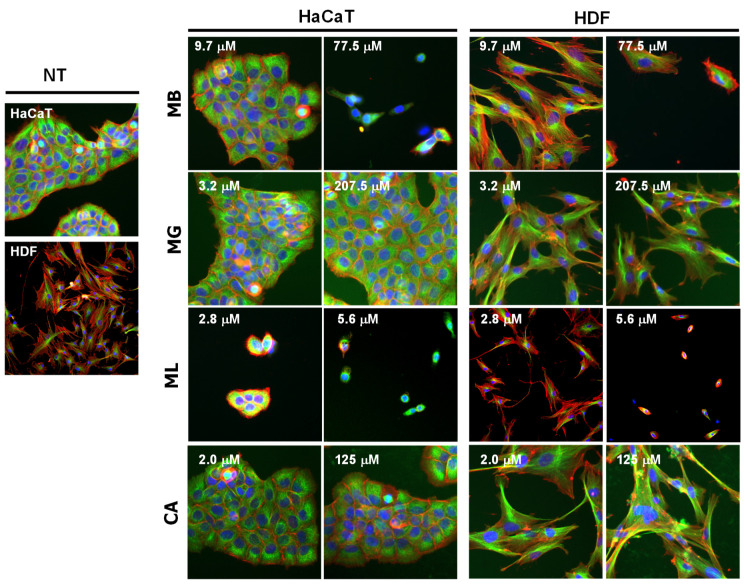
High content imaging showing morphological changes in HaCaT and HDFs exposure to peptides (indicated concentration of MB, MG, ML, and CA) for 24 h. Cells were fluorescently stained for cytoskeleton proteins, α-tubulin (green) and actin (red), and nuclei (blue) to visualize cellular and nuclear morphologies. Cell images were captured using an automated IN Cell Analyzer 2200 microscope. NT indicates no treatment control.

**Figure 3 molecules-27-01438-f003:**
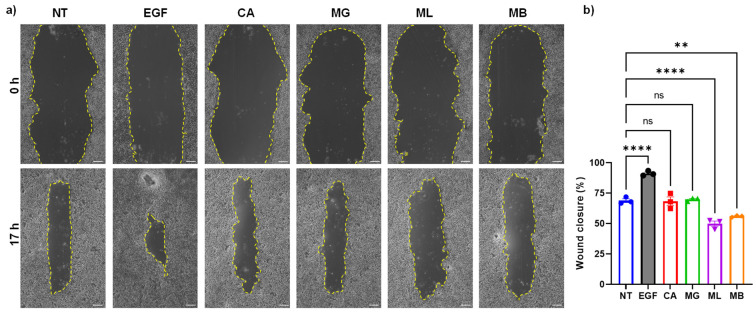
Effect of pore-forming peptides on keratinocyte cell migration. (**a**) Photographs showing the effect of peptides on the migration of NTLG1 cells that were treated with CA (2.0 µM), MG (3.32 µM), ML (2.82 µM), MB (4.96 µM), epidermal growth factor (EGF, 3.1 nM), or untreated (NT). Yellow dotted lines indicate the border of wounds at the indicated timepoint with 4× magnification. Scale bar = 100 μm. (**b**) Quantification of wound closure after 17 h post-addition of peptides or EGF. The data are expressed as mean  ±  SEM (*n* = 3; ns, non-significant ** *p* < 0.01, **** *p* < 0.0001).

**Figure 4 molecules-27-01438-f004:**
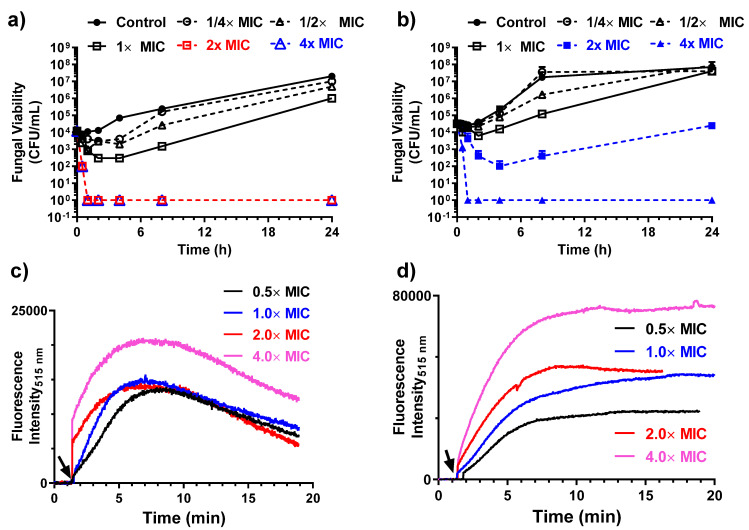
Time–kill kinetics of MB peptides against (**a**) *C. parapsilosis* 1002 and (**b**) *C. glabrata* 1003 strains. Sytox green uptake showing concentration-dependent changes in the dye uptake upon addition of MB to (**c**) *C. parapsilosis* 1002 and (**d**) *C. glabrata* 1003 strains. MIC value of MB against both the strain is 4.96 μM. The black arrow indicates the time of addition of MB peptide. The concentration of peptide is expressed in terms of MIC×.

**Figure 5 molecules-27-01438-f005:**
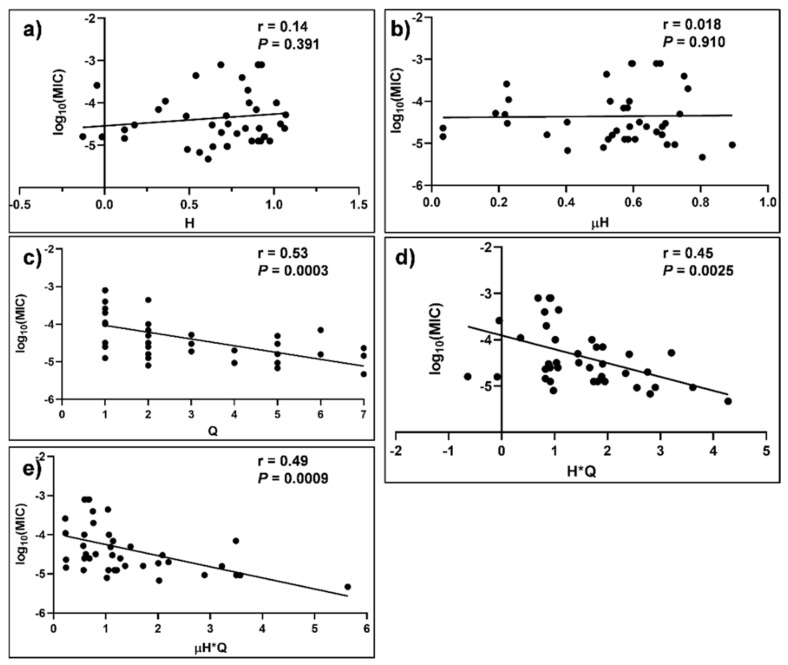
Dependence of MIC of α-helical peptides against *C. glabrata* and various physicochemical parameters. Spearman correlation analysis for MIC is plotted against (**a**) hydrophobicity (H); (**b**) hydrophobic moment (μH); (**c**) overall net charge (Q); (**d**) Hydrophobicity and charge (H*Q) and (**e**) Hydrophobic moment (μH*Q).

**Figure 6 molecules-27-01438-f006:**
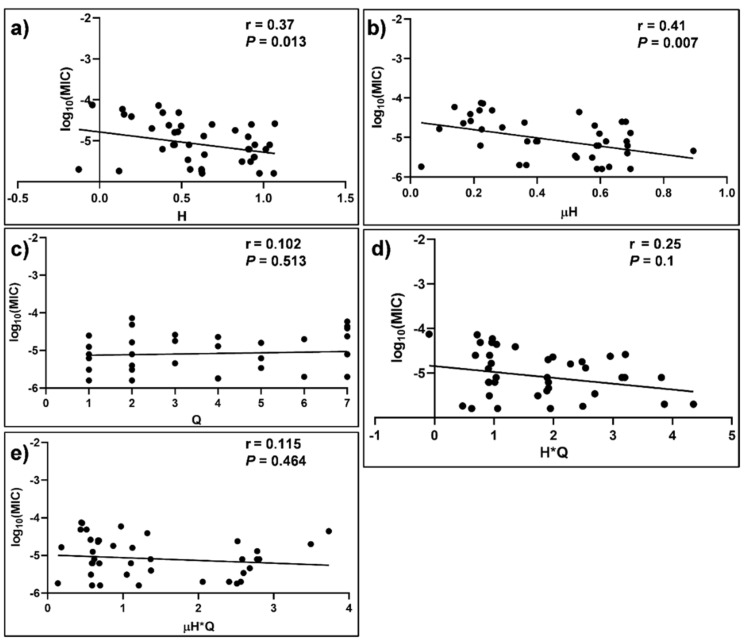
Dependence of MIC of α-helical peptides against *C. tropicalis* and various physicochemical parameters. Spearman correlation analysis for MIC is plotted against (**a**) hydrophobicity (H); (**b**) hydrophobic moment (μH); (**c**) overall net charge (Q); (**d**) hydrophobicity and charge (H*Q) and (**e**) hydrophobic moment and charge (μH*Q).

**Table 1 molecules-27-01438-t001:** Physicochemical properties of prolific pore-forming peptides.

Peptide	Amino Acid Sequence	Net Charge	Hydrophobicity (H)	Hydrophobic Moment (μH)	pI	α-Helicity (%) in		ζ-Potential (mV)
						PBS	TFE	
MB	LKLKSIVSWAKKVL-NH_2_	+5	0.561	0.404	14.0	6.0	38.2	6.8 ± 2.5
MG	GIGKFLHSAGKFGKAFVGEIMKS	+3	0.42	0.515	10.0	1.6	35.2	5.6 ± 1.7
ML	GIGAVLKVLTTGLPALISWIKRKRQQ-NH_2_	+6	0.511	0.394	14.0	5.6	47.5	10.8 ± 0.4
CA	KWKLFKKIEKVGQNIRDGIIKAGPAVAVVGQATQIAK-NH_2_	+7	0.312	0.202	11.4	2.5	18.8	7.8 ± 2.1

**Table 2 molecules-27-01438-t002:** MICs of pore-forming peptides against various “*Candida* spp.” strains. NA indicates no activity.

Strains	Source	MIC of Peptides in μM				
		MB	MG	ML	CA	Amphotericin B
*C. albicans*						
ATCC 10231	-	2.52	26.48	2.82	2.00	0.14
ATCC 2091	-	2.52	26.48	1.41	2.00	0.14
ATCC 24433	-	1.26	13.24	1.41	2.00	0.14
1976R	Colon	0.62	6.63	1.41	0.50	0.54
2672R	Urine	0.62	13.24	1.41	2.00	0.14
*C. tropicalis*						
CT 1001	Blood	0.62	0.82	2.82	2.00	0.54
CT 1002	Blood	0.31	0.82	5.64	1.00	0.54
CT 1003	Blood	1.24	3.31	2.82	4.00	0.54
CT 1004	Blood	0.62	0.82	5.62	1.00	0.54
CT 1005	Blood	0.62	0.82	2.82	1.00	0.54
CT 1006	Blood	1.24	0.82	5.64	1.00	0.54
CT 1007	Blood	1.24	3.31	22.56	4.00	0.54
CT 1008	Blood	0.62	3.31	11.24	4.00	0.54
CT 1009	Blood	0.62	0.82	2.82	1.00	0.54
CT 1010	Blood	0.31	0.82	2.82	1.00	0.54
*C. parapsilosis*						
CP 1001	Peritoneal	2.48	6.62	5.62	NA	1.08
CP 1002	Blood	4.96	6.62	11.24	NA	1.08
CP 1003	Blood	4.96	6.62	11.24	NA	1.08
CP 1004	Others	0.62	3.31	22.48	NA	0.54
CP 1005	Blood	4.96	13.24	5.62	NA	1.08
CP 1006	Blood	4.96	13.24	5.62	NA	1.08
CP 1007	Blood	4.96	13.24	5.62	NA	1.08
CP 1008	Blood	2.48	13.24	>22.48	NA	1.08
CP 1009	Blood	1.24	13.24	11.24	NA	1.08
CP 1010	Blood	4.96	26.48	5.62	NA	1.08
*C. glabrata*						
CG 1001	Blood	9.92	26.48	>22.48	NA	0.27
CG 1002	Pleural fluid	9.92	26.48	>22.48	NA	0.27
CG 1003	Blood	4.96	26.48	>22.48	NA	0.27
CG 1004	Blood	79.40	>26.48	>22.48	NA	0.14
CG 1005	Blood	39.70	>26.48	>22.48	NA	0.27
CG1006	Blood	>79.40	>26.48	>22.48	NA	0.27
CG 1007	Blood	4.96	13.24	11.28	NA	0.27
CG 1008	Blood	2.98	6.62	11.28	NA	0.27
CG 1009	Peritoneal fluid	4.96	26.48	>22.48	NA	0.27
CG 1010	Blood	9.92	26.48	22.48	NA	0.27

**Table 3 molecules-27-01438-t003:** Effect of monovalent and divalent cations on antifungal properties of MB. *C. parapsilosis* 1002 strain was used to ascertain the fungicidal properties of MB.

	Bacterial Viability in CFU/mL	
	No Peptide	MB
SD broth	184,250 ± 50,500	500
+25 mM NaCl	161,000 ± 21,000	3350 ± 250
+50 mM NaCl		8400 ± 1100
+100 mM NaCl		21,900 ± 1500
+150 mM NaCl		65,500 ± 3000
+0.5 mM CaCl_2_	157,000 ± 27,000	900
+1.0 mM CaCl_2_		3100 ± 200
+1.5 mM CaCl_2_		6800 ± 500
+2.0 mM CaCl_2_		5900 ± 200

## Data Availability

All data generated during the current study are included in this article and as Appendix A.

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
