# Peer review of "Rationalisation of Antifungal Properties of α-Helical Pore-Forming Peptide, Mastoparan B"

_molecules, 2022, doi:10.3390/molecules27041438_

Round 1
Reviewer 1 Report
The authors in this manuscript entitled "Rationalisation of Antifungal Properties of α-Helical Pore-forming Peptides" focuses on the antifungal activities, cytocompatibility of four peptides, and the relationship of activities with physicochemical parameters. The results presented here have some values.
Major and minor comments to the authors are as follows:
(1)Are MB, MG, ML and CA all pore-forming peptides? Please indicate this.
(2)Give abbreviation of SD broth as SDB, also for PBS, TKS...
(3)Please give more details on SYTOX green uptake assay, such as volumes of fungal cells and SYTOX Green solution. Please indicate “a” and “b” in Figure S1.
(4)Which data supported the “MG is the most hydrophobic and ...... among the four peptides”? What is the result of “Calculation of physicochemical parameters” in Materials and Methods section? Such as hydrophobicity (H) and hydrophobic moment (μH). Please include this results in Table 1.
(5)Line 179, please insert “.” after “Candida spp”
(6)The concentration in Table 2 is μM, not M. Please correct “ MIC of peptides in M” in Table 2.
(7)Please delete line 178. It was repeat.
(8)Please double check the MIC in Table 2. They should be two-fold serial concentration, for 22.48 and 22.5, 11.24 and 11.2. Please unified them.
(9)Line 186-187, “ML displayed heightened toxicity for both the mammalian cells as complete loss of viability was observed at 1-2× the MIC values (Fig. 1c).”. The MIC here is median MIC or what? For different strains, the MIC is various. Also for other description include MIC.
(10)Is there any significant difference in cytotoxicity of peptide among different tested concentrations? Please include this in result.
(11)The Figure serial numbers of the 3.4. result section is not consistent with that of Figure 3, please correct it.
(12)Line 215, C. parapsilosis should be italic. Also for other fungal species name thoroughly.
(13)Please indicate the strains no. of tested C. parapsilosis and C. glabrata in 3.4. result section.
(14)There is no any result on 3.4 Antifungal activity of MB in the presence of mono and divalent cations, except Table 3. Please describe this result.
(15)Please indicate the MIC value of each peptide in Table S1 and S2. By the way, why the value of log10(MIC) is negative?
(16)This study concluded that the Q, H*Q or H*Q of the peptides could describe the anti-C. glabrata activity. Similarly, H and H of the peptides could describe the anti-C. tropicalis activity of peptides. Could these parameters of the tested 4 peptides in this study describe the different activities? Please discuss/explain this.
Author Response
We thank the reviewer for the comments. We have submitted point-by-point response to the comments/criticisms raised by the reviewer.

Reviewer 2 Report
This paper measured the anti-Candida activity of four synthetic peptides of MB, MG, ML and CA, and the cytocompatibility of the peptides with immortalized human keratinocytes (HaCaT) and primary human dermal fibroblasts (hDF); The more active MB was further analyzed for its Time-kill kinetics against Candida parapsilosis and Candida glabrata. Finally, 85 peptide sequences with anti-Candida activity were found through literature research. The correlation between physicochemical parameters and antifungal was analyzed.. These findings have potential therapeutic implications for the design and development of safe antifungal peptide-based drugs.
The four peptides selected in this paper are relatively mature cationic peptides. There are many errors and imperfections in the paper. Some data are not statistically compared, The author needs to modify it carefully.
Comments and suggestions:
1. The foreword should add a brief introduction to the four peptides used in the text and reasons for selection.
2. Line 83: It is not enough to repeat the measurement of MIC with 2 replicate wells. In addition to the replicate wells, is there any experiment repetition?
3. The source of the clinical strain used in the article should be provided.
4. The name of peptides in Fig1 should be consistent with other Figs, and I suggest that should be changed into abbreviations.
5. SYTOX green uptake assay: the OD in method(line 121) was 520nm, but in the result (Fig3) OD was 515nm.
6. Pleases provided the whole results in Table 2. Because 35 strains of Candida were determined their antifungal activity (line163), but only 32 strains’ MIC were showed in Table 2.
7. The concentration units of the peptide should be showed in Table 2.
8. In line168-169: the MB, ML and CA were effective with a median MIC of 1.24, 1.40 , 1.99, and 13.28.However, the medians actually calculated based on the MIC values in Table 2 are 1.498, 1.682, 1.7 and 17.326 respectively. In line 170: C. tropicalis strains were susceptible to all the four pore-forming peptides with median MIC ranging from 0.62 (MB) to 4.21 (ML). This result was different with the table. Is the author too careless? Or is there a problem with the data? Need to be carefully verified.
9. In line 213: I did not find the any details that antifungal activity of MB in the presence of mono and divalent cations, please add that in Materials and Methods.The description of the results of time sterilization kinetics is not detailed enough.
10.In line 217:The ATP release research mentioned here feels very abrupt. What does this sentence want to show here?
11. The curve of the control group is suggested to be added in Figure 3 a and b. Regarding Table 3, there is no description in the text, please add in the results section.
12. In line 233 Correlation of physicochemical properties of MB with anti-C. glabrata activity: In this part, the relationship between some characteristics of the peptide sequence and the antibacterial activity is discussed, which has certain significance for the subsequent mining of antifungal peptides, but it is not closely related to the other research content of this article. The significance of existence in the article is worth thinking about.
13. The data in Figure 1 and Table 3 are not statistically compared?
Author Response
We thank the reviewer for the overall positive comments. We are providing a point-by-point response to the criticisms of the reviewer.

Reviewer 3 Report
The manuscript by Lim et al. (2021) reports the antifungal activity of four pore-forming alpha-helical AMPs. These peptides were studied regarding their minimal inhibitory concentrations against different pathogenic Candida ssp strains, time-kill kinetics, cytocompatibility, and secondary structure characterization in hydrophilic and hydrophobic environments. Additionally, a physicochemical property/antifungal activities correlation was also presented. In general, the manuscript is well-written, and the technical quality is sound. Nevertheless, after a careful evaluation, I recommend the authors prepare the second version of this manuscript, considering the minor and major corrections described below:
Introduction, 1st paragraph: “spleen which” should read “spleen, which”. The same should be done for “most common AMPs which” (last paragraph – introduction); “concentration which showed” topic 2.1 (material and methods).
Introduction, last paragraph: The authors affirm, "However, their effectiveness against pathogenic and antifungal resistant NACs remains less understood". Please double-check.
By searching for melittin in APD (ID: AP00146) and DADBAASP (ID: 806) databases, the following information was retrieved: Melittin hinders fungal growth (numerous strains, see the ref) by several mechanisms such as membrane permeabilization, apoptosis induction by reactive oxygen species-mediated mitochondria/caspase-dependent pathway, inhibition of (1,3)-beta-D-glucan synthase, and alterations in fungal gene expression (Memariani and Memariani, 2020 - DOI: 10.1007/s00253-020-10701-0).
Please specify which magainin was used (magainin 2?).
Material and methods
Please add the RP-HPLC and mass spec data as supporting information.
Please standardize “ml or mL”.
Topic 2.1: “37 ºC” should read “37 oC”. This same issue is found in other sections of the MS. Please double-check.
Considering that the study was carried out with four different pore-forming AMPs of distinct molecular weights (ranging from 1613.06 Da to 4004.82 Da – theoretical MW), why did the authors perform the MIC assay in microgram/mL? The values should be presented in mmol L-1. Please update accordingly. If the data is presented in microgram/mL the authors must inform the MWs and affirm that the molar concentration is higher for the peptide with the lowest MW (mastoparan B).
- Just saw that Table 2 is in uM. All good. Just update the concentration range to uM in the material and methods.
Topic 2.2: The authors affirm they used human fibroblast to evaluate the cytocompatibility of each peptide studied. Candida infections are commonly reported in cutaneous wounds. Bearing this in mind, would the authors consider performing a wound scratch assay in microplates to evaluate the potential of each peptide in stimulating fibroblasts migration (wound healing)? It is a standard experiment and that would enhance the manuscript’s quality. For more information, see: Veeraraghavan, V.P. et al. Green synthesis of silver nanoparticles from aqueous extract of Scutellaria barbata and coating on the cotton fabric for antimicrobial applications and wound healing activity in fibroblast cells (L929). Saudi J Biol Sci 28, 3633-3640 (2021).
Topic 2.4: “trifluoroethanol"should read “2,2,2-trifluoroethanol”. Please indicate the TFE percentage (e.g., 30% (v/v) in water or PBS?). Please add the peptides’ concentration. Please indicate the temperature (room temperature?). PBS concentration (10 mM??)? TFE is commonly used to study peptides' secondary structure as it displaces water molecules around the peptide backbone, favoring folding. Nevertheless, it does not accurately mimic a fungal cell wall/membrane. Therefore, I highly recommend that the authors perform additional CD experiments using large unilamellar vesicles (LUVs) mimicking fungal cells surfaces (e.g., adding ergosterol to a zwitterionic LUV – POPC/Erg). For more information, please see: 10.1016/j.bbamem.2018.12.020. I also recommend less complex systems, including SDS and DPC micelles, to mimic anionic and zwitterionic bilayers.
Topic 2.6.: “means of the Heliquest” software (http://heliquest.ipmc.cnrs.fr/)”. Heliquest is a web server, not software. Please double-check.
Results: The results are not presented in the same order reported in the material and methods. Please check.
Topic 3.1: CD is used to characterize the secondary structure of biomolecules, not to determine peptide structures as claimed by the authors. Structure determination of short linear peptides is usually obtained through solution NMR. How was the helicity (%) calculated? The zeta-potential experiments lack more relevant analyses. For instance, the authors should consider the evaluation of zeta-potentials for membrane bilayers mimicking fungi cells surfaces (e.g., POPC/Ergosterol) to shed light on the peptides' influence on membrane depolarization.
In general, the structural data is not well-explored. The authors should consider improving this section of the MS.
FigureS1: please double-check "poreforming". It should read "pore-forming". Why did the authors use 200 uM of each peptide for CD analysis? It is quite high than usually reported (from 30 to 50 uM in most cases).
Figure 1: please, insert all numbers in the y-axis, not only 120%. Figure 1d, black bar, 3.9 uM, the SD bar is cut at the top. Please adjust all the graphs, so the font sizes are the same. Transfer (a), (b) … etc, from the right to the left side of the graphs.
Figure 1 caption: “Note that MB and ML decreased the metabolic activity of the cells at elevated concentration”. It can be said for MB only. ML decreases the cell viability at low concentrations (starting from 2.8 uM).
Figure 3. Please double-check the font size for all the graphs, they are not standardized.
The authors should consider performing additional experiments to evaluate reactive oxygen species production in the presence of the four peptides tested. It is commonly reported in studies reporting novel antifungal activities for a given molecule.
Topic 3.5. These data are confusing. Why did the authors select only MB? MB is a 14-amino acid residues peptide. Therefore, why did the authors decide for AMPs with antifungal properties and containing <=18 amino acid residues? Where did these sequences come from? Antimicrobial peptide database? This methodology is not in the material and methods. Please add. The authors should consider a more refined analysis. First, it would be better to choose a database. If the authors are investigating MB only (which is also not explained why), it would be better to retrieve peptide sequences with 14-amino acid residues. After that, another thing that should have been done is to submit all the sequences obtained to a pattern identification server (e.g., PRATT 2.1). By doing this, the authors would determine a given amino acids pattern for alpha-helical, 14-amino acid residues antifungal peptides. In this topic, it is also unclear whether the MB data is shown in the graphs. Bearing this in mind, the supplementary tables must be updated accordingly.
Please add a separate paragraph for the conclusion after the discussion section.
The title does not correspond to the data presented. There is not "rationalization" in this MS. Please update it.
Author Response
We thank the reviewer for the comments. A point-by-point response is provided in the attached file.

Round 2
Reviewer 1 Report
Based on this revised MS, I don't have more comments.
Reviewer 3 Report
The authors have addressed all my suggestions accordingly, improving the manuscript. My final recommendation is to accept the manuscript in its present form.